# Plant-Produced S1 Subunit Protein of SARS-CoV-2 Elicits Immunogenic Responses in Mice

**DOI:** 10.3390/vaccines10111961

**Published:** 2022-11-18

**Authors:** Chalisa Panapitakkul, Narach Khorattanakulchai, Kaewta Rattanapisit, Theerakarn Srisangsung, Balamurugan Shanmugaraj, Supranee Buranapraditkun, Chutitorn Ketloy, Eakachai Prompetchara, Waranyoo Phoolcharoen

**Affiliations:** 1Center of Excellence in Plant-Produced Pharmaceuticals, Chulalongkorn University, Bangkok 10330, Thailand; 2Department of Pharmacognosy and Pharmaceutical Botany, Faculty of Pharmaceutical Sciences, Chulalongkorn University, Bangkok 10330, Thailand; 3Baiya Phytopharm Co., Ltd., Bangkok 10330, Thailand; 4Center of Excellence in Vaccine Research and Development (Chula VRC), Chulalongkorn University, Bangkok 10330, Thailand; 5Division of Allergy and Clinical Immunology, Department of Medicine, King Chulalongkorn Memorial Hospital, Faculty of Medicine, Chulalongkorn University, Thai Red Cross Society, Bangkok 10330, Thailand; 6Thai Pediatric Gastroenterology, Hepatology and Immunology (TPGHAI) Research Unit, Faculty of Medicine, Chulalongkorn University, Bangkok 10330, Thailand; 7Department of Laboratory Medicine, Faculty of Medicine, Chulalongkorn University, Bangkok 10330, Thailand

**Keywords:** SARS-CoV-2, subunit vaccine, spike protein, plant-produced recombinant protein

## Abstract

SARS-CoV-2 is responsible for the ongoing COVID-19 pandemic. The virus spreads rapidly with a high transmission rate among humans, and hence virus management has been challenging owing to finding specific therapies or vaccinations. Hence, an effective, low-cost vaccine is urgently required. In this study, the immunogenicity of the plant-produced S1 subunit protein of SARS-CoV-2 was examined in order to assess it as a potential candidate for SARS-CoV-2. The SARS-CoV-2 S1-Fc fusion protein was transiently produced in *Nicotiana benthamiana*. Within four days of infiltration, the SARS-CoV-2 S1-Fc protein was expressed in high quantities, and using protein A affinity column chromatography, plant-produced S1-Fc protein was purified from the crude extracts. The characterization of plant-produced S1-Fc protein was analyzed by SDS-PAGE and Western blotting. Immunogenicity of the purified S1-Fc protein formulated with alum induced both RBD specific antibodies and T cell immune responses in mice. These preliminary results indicated that the plant-produced S1 protein is immunogenic in mice.

## 1. Introduction

After the first outbreak in Wuhan, China, the severe acute respiratory syndrome coronavirus 2 (SARS-CoV-2) quickly spread throughout the world. [1]. SARS-CoV-2, a *Betacoronavirus* [2,3], has a 79.5% similar genome sequence to its related severe acute respiratory syndrome coronavirus 1 (SARS-CoV-1) [4,5]. SARS-CoV-2 is a single-strand RNA-enveloped virus belonging to the Coronaviridae family [4,6,7]. Currently, few vaccines to protect against SARS-CoV-2 are available, and many are in development, such as mRNA vaccines, DNA vaccines, subunit vaccines, viral vector vaccines, and inactivated virus vaccines [8,9,10]. SARS-CoV-2 has four primary structural proteins: the spike (S) surface glycoprotein, the membrane (M) protein, the envelope (E) protein, and the nucleocapsid (N) protein. [11]. The most important target for developing a vaccine is the spike protein of SARS-CoV-2, as it is involved in the virus attachment and fusion into the host cell [11,12]. The spike protein has a trimeric form, which contains two parts of the S1 and S2 subunits [11]. The S1 subunit includes two domains: the C-terminal (CTD) and the N-terminal (NTD), the latter of which features a receptor binding domain (RBD) that interacts with the host cell receptor by attaching with angiotensin-converting enzyme 2 (ACE2) [11,12]. The majority of the epitopes that the neutralizing antibodies target are found on the spike protein, especially on the S1 subunit [13] and RBD domain [14,15,16], which can be considered as the major targets for vaccine development.

Due to the rapid spread of SARS-CoV-2 and its variants, vaccines that are both safe and broadly effective against variants are urgently needed. Therefore, it is necessary to select an appropriate expression system for subunit antigen production [17,18]. The available expression system for recombinant protein synthesis are mammalian cells, insect cells, or yeast, but every system also has certain limitations [19]. Hence, this study utilizes plants to produce the S1 subunit of SARS-CoV-2 due to its advantages, such as cost-effectiveness, speed, and scalability, which were reviewed in much of the literature [19,20,21]. For these reasons, this study used *Nicotiana benthamiana* to produce the recombinant SARS-CoV-2 S1-Fc protein by combining the SARS-CoV-2 S1 with the Fc domain of human immunoglobulin G1 (IgG1). Briefly, SARS-CoV-2 S1-Fc was cloned and expressed in plants using a geminiviral expression vector. The SARS-CoV-2 S1-Fc protein produced by plants was characterized, purified, quantified, and its immunogenicity was tested in mice. 

## 2. Materials and Methods

### 2.1. Construction of Plant Expression Vector for SARS-CoV-2 S1-Fc

The SARS-CoV-2 S1 (Genbank accession number: YP_009724390.1) was modified to include a peptide linker at the C-terminus that allows it to anneal with the Fc region of human immunoglobulin G1 (IgG1) (GenBank accession number: 4CDH_A). The SARS-CoV-2 S1 nucleotide sequence was amplified from the total DNA S1-His plasmid by using the polymerase chain reaction (PCR) with specific primers pairs (*Xba*I-SP F 5′ TCT AGA ACA ATG GGC TGG 3′ and *Bam*HI-GS R 5′ CGG GAT CCA CCA CCA CCA GAG ATA TCT CTA GCC CTT CTA GGA G 3′ (Bionics, Gwangju, South Korea). To generate the SARS-CoV-2 S1-Fc geminiviral expression vector pBYR2eK2Md (pBYR2e), the human Fc region was digested with *BamH*I/*Sac*I and ligated into the geminiviral vector containing the SARS-CoV-2 S1 for plant expression (Figure 1).

### 2.2. SARS-CoV-2 S1-Fc Expression in N. benthamiana

The construct pBYR2e-SARS-CoV-2 S1-Fc was electroporated into *Agrobacterium tumefaciens* GV3101 using the MicroPulser (Bio-Rad, Hercules, CA, USA). The recombinant *A. tumefaciens* clones were identified and validated using colony PCR with S1 gene primers (*Xho*I S-plant-F 5′-GGG CTC GAG GGG ATG TTC GTG TTC CTT GTG CTG CTT CCG CTT GTG TCA TCT CAG TGC G-3′ and *BamH*I-GS R 5′-CGG GAT CCA CCA CCA CCA GAG ATA TCT CTA GCC CTT CTA GGA G-3′). *A. tumefaciens* harboring pBYR2e-SARS-CoV-2 S1-Fc was combined with 1x infiltration buffer (10 mM 2-N-morpholino-ethanesulfonic acid (MES) and 10 mM MgSO_4_, pH 5.5) to obtain an optimal density at 600 nm (OD_600_) of 0.2 for vacuum infiltration of the *Agrobacterium* suspension into the underside of *N. benthamiana* leaves. The infiltrated *N. benthamiana* leaves were kept under controlled conditions in an indoor plant room. To measure the SARS-CoV-2 S1-Fc protein expression, the leaves were collected at 2-, 4-, 6-, 8-, and 10-days post infiltration (dpi). Leaves were ground using 1× PBS buffer (137 mM NaCl, 2.7 mM KCl, 10 mM Na_2_HPO_4_, and 1.8 mM KH_2_PO_4_) to extract the plant-produced SARS-CoV-2 S1-Fc protein. The plant-produced SARS-CoV-2 S1-Fc protein was quantified by ELISA in a 96-well plate (Greiner Bio-One, Frickenhausen, Germany) coated with 50 µL of plant-produced SARS-CoV-2 S1-Fc protein. Each sample was diluted (1:100, 1:200, 1:400, and 1:800) in 1× PBS and incubated overnight 4 °C. The 96-well plate was washed thrice with PBST buffer (0.05% Tween 20 in 1× PBS) and blocked with 5% skim milk (BD Difco, New Jersey, USA) in 1× PBS for 2 h at 37 °C. After the blocking and washing step, the 96-well plate was incubated with horseradish peroxidase (HRP)-conjugated goat anti-human IgG (Southern Biotech, Birmingham, AL, USA) diluted (1:2000) in 3% skim milk for 1 h at 37 °C. Finally, the 96-well plate was washed thrice with PBST buffer before being developed using a TMB One Solution (Promega, Madison, WI, USA) and incubated for at least 2 min. The reaction will be stopped by 50 µL of 1M H_2_SO_4_ and measured at 450 nm with a 96-well plate reader (BMG Labtech, Ortenberg, Germany). All data were calculated by using the mean ± SD of triplicates.

### 2.3. Purification of SARS-CoV-2 S1-Fc Protein

The plant-produced SARS-CoV-2 S1-Fc protein was purified, as previously mentioned [22]. The purified plant-produced SARS-CoV-2 S1-Fc was concentrated with Amicon^®^ ultra centrifugal filter (Merck, NJ, USA) and filtered through a 0.22 µm syringe filter (Merck, NJ, USA). The purified SARS-CoV-2 S1-Fc protein was examined by sodium dodecyl sulfate-polyacrylamide gel electrophoresis (SDS-PAGE) and Western blotting [23]. Briefly, the purified SARS-CoV-2 S1- Fc samples were combined with non-reducing loading buffer (125 mM Tris-HCl pH 6.8, 12% (*w*/*v*) SDS, 10% (*v*/*v*) glycerol, 0.001% (*w*/*v*) bromophenol blue) and reducing loading buffer (125 mM Tris-HCl pH 6.8, 12% (*w*/*v*) SDS, 10% (*v*/*v*) glycerol, 22% (*v*/*v*) β-mercaptoethanol, 0.001% (*w*/*v*) bromophenol blue) before being separated on 4–15% SDS-PAGE (Bio-Rad, Hercules, CA, USA) and observed with a staining solution named InstantBlue^TM^ (Abcam, Cambridge, UK). For Western blot analysis, the protein gel that had separated was transferred to a nitrocellulose membrane (Bio-Rad, CA, USA) and horseradish peroxidase (HRP)-conjugated goat anti-human IgG (Southern Biotech, Birmingham, AL, USA) diluted (1:10,000) in 3% skim milk was probed before detection with enhanced chemiluminescence (ECL) plus detection reagent (Thermo Fisher Scientific, Waltham, MA, USA) before exposed with Chemiluminescent ImageQuant^TM^ LAS500 (GE Healthcare Bio-Sciences AB, Uppsala, Sweden).

### 2.4. Quantification of Plant-Produced SARS-CoV-2 S1-Fc Protein

The yield of purified plant-produced SARS-CoV-2 S1-Fc protein was determined by ELISA. Briefly, 96-well microplate (Greiner Bio-One, Frickenhausen, Germany) was coated with 2 µg/mL monoclonal antibody H4 diluted in 1× PBS buffer and incubated overnight at 4 °C. The 96-well microplate was washed three times with PBST buffer (0.05% Tween 20 in 1× PBS). Then the 96-well microplate was blocked with 5% skim milk (BD Difco, NJ, USA) in 1× PBS for 2 h at 37 °C. After the blocking and washing step, the SARS-CoV-2 spike protein RBD (GenScript, Piscataway, NJ, USA) and samples were added into the microplate and incubated for 2 h at 37 °C. Then the plate was washed and incubated with SARS-CoV-2 Spike antibody (HRP) (Sino Biological, Beijing, China) diluted 1:1000 in 1× PBS for 1 h at 37 °C. Finally, the plate was washed before being developed using a TMB One Solution (Promega, Madison, WI, USA) and incubated for at least 2 min. The reaction was terminated with 50 µL of 1M sulfuric acid (H_2_SO_4_), and the absorbance at 450 nm was measured using a 96-well plate reader (BMG Labtech, Ortenberg, Germany).

### 2.5. Mice Immunization with the SARS-CoV-2 S1-Fc Protein

The protocol of mice immunization was endorsed by the Institutional Animal Care and Use Committee, Faculty of Medicine, Chulalongkorn University (Protocol No. 013/2564). On days 0 and 21, four-week-old female ICR mice (*n* = 5 per group) were intramuscularly (IM) injected with 10 µg of plant-produced SARS-CoV-2 S1-Fc protein formulated with 0.1 mg aluminum hydroxide gel adjuvant (Croda, Frederikssund, Denmark). To evaluate the SARS-CoV-2-specific antibody response, mice sera were obtained before the first inoculation (baseline) and 14 days after each administration. The mice were euthanized 14 days following the second immunization (day 35), and splenocytes were collected for investigation of SARS-CoV-2 RBD-specific T-cell responses.

### 2.6. Evaluation of Immunological Responses in Mice

SARS-CoV-2 RBD specific total antibody responses were determined using Sf9 insect cells SARS-CoV-2 spike protein RBD (GenScript, NJ, USA) as coating antigen and goat anti-mouse IgG HRP conjugate antibody (Jackson ImmunoResearch, Westgrove, PA, USA) as the detection antibody. The endpoint titers were determined by following the method, as described previously [16]. The cells secreting mouse IFN-γ by SARS-CoV-2-specific cells were evaluated using an ELISpot test for mouse IFN-γ (Mabtech, Stockholm, Sweden) by following the protocol as described previously [16], and the results are given in terms of spot-forming cells (SFCs)/10^6^ splenocytes. 

### 2.7. Statistical Analysis 

The statistical analyses were carried out using GraphPad Prism 9.0 (GraphPad Software, Inc., CA, USA). To calculate the results of total IgG and IgG subclasses, two-way analysis of variance (ANOVA), Tukey test, and multiple comparison tests were used. Mann–Whitney test was used to calculate the IFN-γ ELISpot assay results. All *p* values < 0.05 were defined as significant.

## 3. Results

### 3.1. SARS-CoV-2 S1-Fc Expression in N. benthamiana

The SARS-CoV-2 S1-Fc fusion protein was cloned into the geminiviral plant expression vector named pBYR2e. *Agrobacterium* harboring pBYR2e-SARS-CoV-2 S1-Fc was agroinfiltrated into *N. benthamiana* plants (Figure 1). The plants infiltrated with *Agrobacterium* containing pBYR2e-SARS-CoV-2 S1-Fc construct exhibited necrosis in comparison to control plants (Figure 2a). For the time-course experiment, at 2, 4, 6, 8, and 10 dpi, the infiltrated leaves were collected. The yield of SARS-CoV-2 S1-Fc protein was quantified by ELISA. The optimal expression of SARS-CoV-2 S1-Fc protein was obtained four days after infiltration, and the protein accumulated up to 30 µg/g fresh leaf weight (Figure 2b).

### 3.2. Purification and Characterization of SARS-CoV-2 S1-Fc from N. Benthamiana Leaves

The plant-produced SARS-CoV-2 S1-Fc was purified from a crude extract of *N. benthamiana* leaves using single-step protein A affinity chromatography. The SARS-CoV-2 S1-Fc purified from plants was concentrated and filtered using a 0.22 µm syringe filter (Pall Corporation, NY, USA). The purified SARS-CoV-2 S1-Fc protein was characterized by SDS-PAGE and Western blot analysis. The expected band at the size of 100–150 kDa and 250 kDa was observed under reducing (Figure 2c; Lane 1) and non-reducing conditions in the InstantBlue-stained SDS gel (Figure 2c; Lane 2). Western blot analysis with anti-human gamma chain-HRP conjugated antibody confirmed the molecular weight of SARS-CoV-2 S1-Fc at 100–150 kDa and 250 kDa under reducing and non-reducing conditions, respectively (Figure 2d; Lane 1 and Lane 2; Appendix A). The yield of plant-produced SARS-CoV-2 S1-Fc protein was measured using ELISA and determined to be 3.9 mg/mL.

### 3.3. Immunogenicity in Mice

On days 0 and 21, mice were immunized with 10 µg of plant-produced SARS-CoV-2 S1-Fc with alum adjuvant, and sera were collected on days 0, 14, and 35, as shown in Figure 3a. The evaluation of SARS-CoV-2 RBD-specific antibodies was analyzed by ELISA using commercial Sf9-produced SARS-CoV-2 RBD-His as a capture antigen. To measure IgG response, mice were immunized with 10 µg of plant-produced SARS-CoV-2 S1-Fc with alum or alum alone as a control. The sera were collected on days 0, 14, and 35. The SARS-CoV-2 RBD-specific total IgG of S1-Fc immunized mice was detected after 14 days of second immunization (geometric mean end-point titer (GMT) = 919) which was significantly higher than the control group with *p* < 0.01, but not at 14 days after first immunization (Figure 3b). Moreover, the mice sera were also subjected to IgG subtypes evaluation. The RBD-specific-IgG1 (Figure 3c) and -IgG2a (Figure 3d) titers, induced by the SARS-CoV-2 S1-Fc protein, were not significantly different compared with the control group. However, but RBD-specific-IgG1 titers (GMT = 1600), was found to be higher than SARS-CoV-2 RBD-specific IgG2a titers (GMT = 459.47) with a IgG1/IgG2a ratio of 3.38 folds (Figure 3e).

### 3.4. IFN-γ ELISpot Assay

The immunized mice splenocytes were isolated and evaluated for RBD-specific IFN-γ secretion using IFN-γ ELISpot assay on day 35. The findings showed that compared to the control group, the plant-produced SARS-CoV-2 S1-Fc considerably increased the IFN-γ secretion with *p* < 0.01 (Figure 4, Appendix A).

## 4. Discussion

Since the emergence of SARS-CoV-2, researchers have been working towards the development of vaccines in order to decrease mortality and morbidity [12,24]. Earlier studies indicated that the SARS-CoV-2 spike protein can induce a potent immune response and neutralize antibodies, and it is considered suitable for the development of a recombinant vaccine [12,25,26]. The S protein is found on the surface of the virus particles involved in the virus entry into the host cell [12]. However, this study is interested in the S1 subunit because the S1 subunit has considerable neutralization epitopes than the S2 subunit [13]. Moreover, the receptor binding domain (RBD) from the S1 subunit induced significant immune response and neutralizing antibodies which were evident from several studies [14,15]. The full length of the S1 subunit also can stimulate the immune response and the neutralization antibodies to block the viral infection [27,28,29]. As a result, the S1 subunit in the S protein could be a target for developing an effective SARS-CoV-2 vaccine.

The number of COVID-19 cases is rapidly increasing; hence, a platform that can produce antigens in a short time to make the recombinant vaccine is required. Plants are a suitable platform to use during an emergency situation because, in comparison to other expression systems, it has many advantages, such as cost, safety, and speed of production, and it has been largely used for producing biopharmaceuticals over the past decade [20,30,31,32,33]. Many research groups have evaluated and thoroughly studied the feasibility of plant-derived biopharmaceuticals and vaccines [34,35]. In particular, plants are used as an expression system for antibodies, vaccines, diagnostic reagents, human serum albumin, cytokines, and growth factors production [36]. The capacity of plants to produce recombinant biopharmaceuticals against SARS-CoV-2 has also been documented [37,38,39,40,41,42,43]. Notably, the plant-derived biopharmaceutical taliglucerase alfa (ELELYSO^TM^), used to treat Gaucher disease in adults, was approved by Food and Drug Administration (FDA) [44].

Our study demonstrated the expression of SARS-CoV-2 S1-Fc protein in *N. benthamiana* plants and tested the efficacy in mice. The results showed that the SARS-CoV-2 S1 subunit fused with Fc domain can be expressed in *N. benthamiana* plants with high yield obtained within four days after infiltration. The infiltrated leaves showed necrosis compared to the control. Furthermore, the characterization of plant-produced SARS-S1-Fc protein indicated that the expected size of approximately 100–150 kDa and 250 kDa in reducing and non-reducing conditions, respectively, was observed in Western blot. In addition, the results depicted that some of the proteins are degraded during processing, and hence, further optimization is required to enhance protein accumulation and prevent protein degradation during downstream processing.

The plant-produced SARS-CoV-2 S1-Fc protein was prepared using alum as an adjuvant, and the findings indicated the S1 protein induces an antigen-specific antibody response. Previous report showed that the SARS-CoV-2 S1 subunit protein can stimulate the SARS-CoV-2 S1-specific IgG titer higher than the SARS-CoV-2 RBD protein when using the SARS-CoV-2 S1 protein and SARS-CoV-2 RBD protein as a coating antigen in ELISA [28]. Further, alum adjuvant can enhance the Th-2 (IgG1) immune response better than the Th-1 (IgG2a) immune response [45], which confirmed in the present study that the SARS-CoV-2 RBD-specific IgG1 antibody responses were increased more than the SARS-CoV-2 RBD-specific IgG2a antibody responses [27,45].

Nevertheless, RBD-specific IFN-γ secretion analyzed by ELISpot assay indicated that the plant-produced SARS-CoV-2 S1-Fc with alum was immunogenic in mice, as shown in Figure 4, and it has a significant IFN-γ secretion higher than the control group. However, additional experiments, such as neutralizing antibody analysis and viral challenge experiments, are warranted to support these results.

In summary, our research revealed that the S1 subunit of SARS-CoV-2 fused with Fc region can be produced in plants as an expression system with a high yield and can be obtained within four dpi. Further, the plant-produced vaccine-induced antigen-specific antibodies and T-cell responses in mice. These findings provide a basis to further advance the development of a plant-produced subunit SARS-CoV-2 vaccine. 

## Figures and Tables

**Figure 1 vaccines-10-01961-f001:**
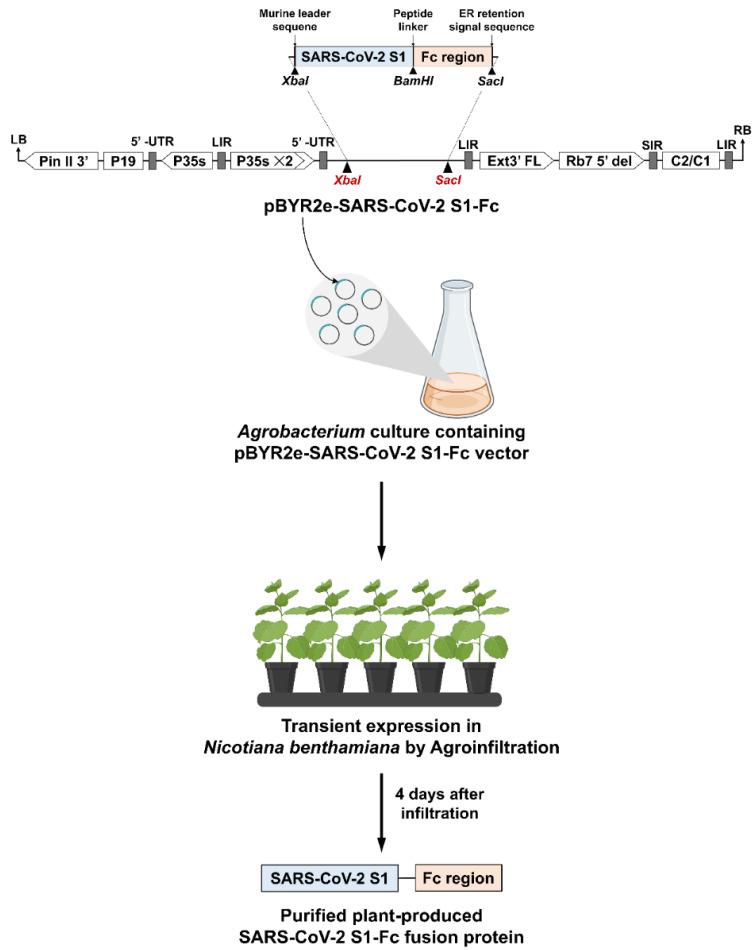
Schematic illustration shows the T-DNA of pBYR2e-SARS-CoV-2 S1-Fc plant expression vector and the *N. benthamiana* plants’ transient expression in brief. The left and right margins of the T-DNA employed in the transfer of *Agrobacterium* DNA into plant cells, RB and LB; as a terminator, the potato proteinase inhibitor II gene was used, Pin II 3′; the Tomato Bushy Stunt Virus (TBSV) as an RNA silencing suppressor, P19; the Cauliflower Mosaic Virus (CaMV) serves as a 35s promoter, P35s; the CaMV with duplicated enhancer as a 35s promoter, P35s × 2; the region of tobacco extension gene, Ext3′ FL, 3′; the tobacco RB7 promoter, Rb7 5′ del; the Bean Yellow Dwarf Virus (BeYDV) short intergenic region, SIR; the BeYDV long intergenic region, LIR; the BeYDV encoding for replication initiation protein (Rep) and RepA as a Rep/RepA gene, C2/C1.

**Figure 2 vaccines-10-01961-f002:**
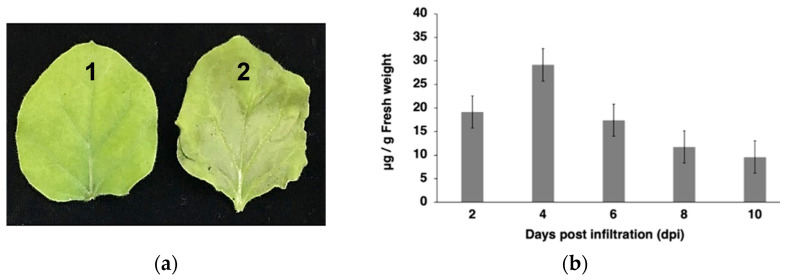
Expression and characterization of plant-produced SARS-CoV-2 S1-Fc protein. (**a**) The phenotype of wild-type leaves (1) and *Agrobacterium*-infiltrated leaves with pBYR2e-SARS-CoV-2 S1-Fc after 4 dpi (2). (**b**) Optimization of SARS-CoV-2 S1-Fc protein expression in *N. benthamiana* leaves performed on days 2-, 4-, 6-, 8-, and 10 after agroinfiltration using ELISA. The data are provided as the mean ± SD of three replicates. (**c**) SDS-PAGE analysis of purified SARS-CoV-2 S1-Fc protein produced in plants stained with InstantBlue™ and (**d**) Western blot analysis of purified plant-produced SARS-CoV-2 S1-Fc protein probed with anti-human gamma-HRP conjugate antibody. The SARS-CoV-2-S1-Fc protein purified from plants was loaded in Lane 1 and 2 under reducing and non-reducing conditions, respectively.

**Figure 3 vaccines-10-01961-f003:**
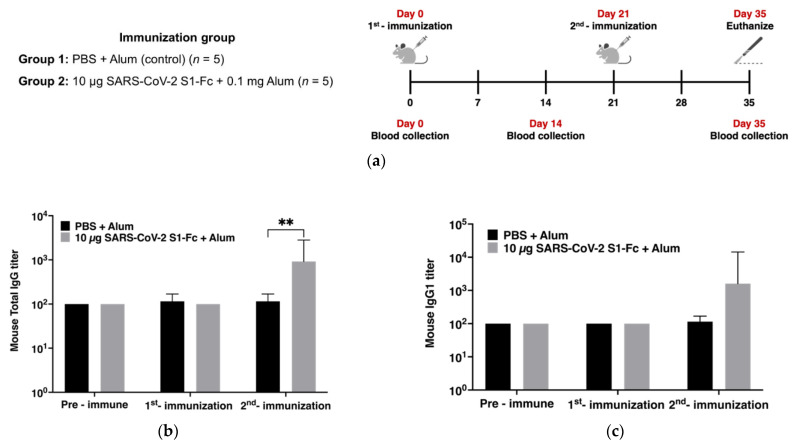
Mice immunogenicity studies. Diagrammatic illustration of immunization procedure and sample collection. Groups of mice (*n* = 5) were immunized with 10 µg of SARS-CoV-2 S1-Fc protein adjuvanted with alum on days 0 and 21. Mice sera were obtained on days 0, 14, and 35 (**a**). Titers of SARS-CoV-2 RBD-specific mouse (**b**) total IgG, (**c**) IgG1, (**d**) IgG2a, and (**e**) the ratio of SARS-CoV-2 RBD-specific mouse IgG1/IgG2a titer at indicated time-point. Data presented as geometric mean ± 95% CI of the endpoint titers in each group, *n* = 5. Two-way ANOVA, Tukey test, was used (**: *p* < 0.01).

**Figure 4 vaccines-10-01961-f004:**
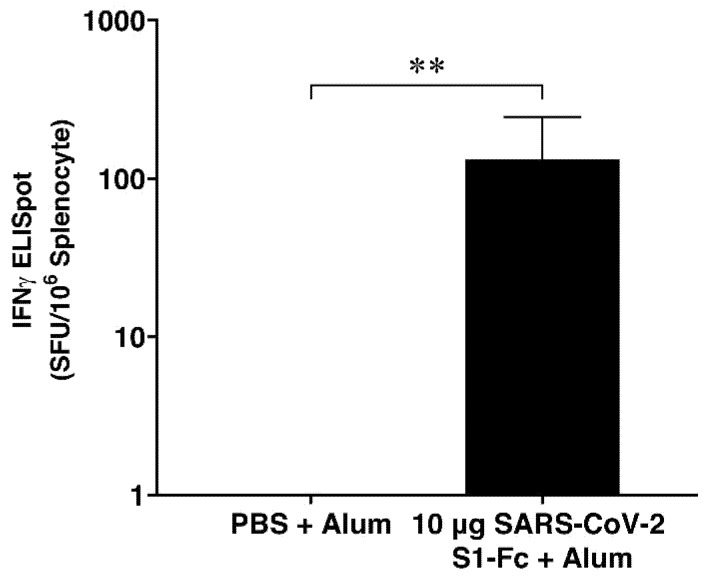
SARS-CoV-2 RBD-specific T-cell responses in immunized mice. Mouse splenocytes were stimulated with RBD peptide pools and analyzed by mouse IFN-γ ELISpot assay. Data presented as mean ± SD (*n* = 5). Mann–Whitney test was used compared with control (**: *p* < 0.01). SFCs: Spot-forming cells.

## Data Availability

All data supporting the findings of this study are available from the corresponding author upon request.

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
