# Peer review of "Plant-Produced S1 Subunit Protein of SARS-CoV-2 Elicits Immunogenic Responses in Mice"

_vaccines, 2022, doi:10.3390/vaccines10111961_

Round 1
Reviewer 1 Report
Specify the amount of total protein loaded on the gel for electrophoresis and Western blotting.
A little is commented on the quality and purity of the expressed fusion protein. Impurities are observed and that could affect the specificity of the antibodies produced. Review and to consider it.
Discuss at length how the use of S1 may represent an advantage over existing vaccines considering the SARS-CoV-2 mutation rate and the new variants.
Author Response
- Specify the amount of total protein loaded on the gel for electrophoresis and Western blotting.
Author’s response: For the electrophoresis, the SARS-CoV-2 S1-Fc protein was loaded on the gel with the concentration of 0.3 µg/well.
For the western blot, the SARS-CoV-2 S1-Fc protein was loaded on the gel with the concentration of 5 µg/well and 1 µg/well for western blot analysis in reducing and non-reducing conditions, respectively. (Supplement Figure S1).
- A little is commented on the quality and purity of the expressed fusion protein. Impurities are observed and that could affect the specificity of the antibodies produced.
Author’s response: We agree with the reviewer. Some of the proteins could be degraded during protein processing. Currently we are optimizing the downstream purification procedures to increase the purity of the expressed proteins in plants.
- Discuss at length how the use of S1 may represent an advantage over existing vaccines considering the SARS-CoV-2 mutation rate and the new variants.
Author’s response: The emergence of SARS-CoV-2 variants led to an urgent demand for a broadly effective vaccine against the threat of variant infection. Based on SARS/MERS experiences, the research vaccine development proposes S1/S2 protein subunits, receptor-binding domain (RBD), and S protein/gene as the most preferred target sites. In addition, recombinant protein vaccines have several advantages: no risk of genome integration, an adequate safety profile, suitable for people with compromised immune systems, high productivity, and good stability. Reports showed that the S1-based vaccines could elicit a comprehensive humoral response toward different SARS-CoV-2 variants of concern and variants of interest and will be helpful to combat COVID-19 globally (Sun et al., 2022; Lin et alk., 2021; Wang et al., 2021).
References:
Sun C, Yuan RY, Xie C, Sun JF, Fang XY, Hu YS, Yu XH, Liu Z, Zeng MS, Kang YF. Induction of Broadly Cross-Reactive Antibody Responses to SARS-CoV-2 Variants by S1 Nanoparticle Vaccines. J Virol. 2022 Jul 13;96(13):e0038322.
Lin HT, Chen CC, Chiao DJ, Chang TY, Chen XA, Young JJ, Kuo SC. Nanoparticular CpG-adjuvanted SARS-CoV-2 S1 protein elicits broadly neutralizing and Th1-biased immunoreactivity in mice. Int J Biol Macromol. 2021 Dec 15;193(Pt B):1885-1897.
Wang Y, Wang L, Cao H, Liu C. SARS-CoV-2 S1 is superior to the RBD as a COVID-19 subunit vaccine antigen. J Med Virol. 2021 Feb;93(2):892-898.
Reviewer 2 Report
1. Can authors explain why there is no IgG detected after the first immunization? I agree that the booster is necessary but at least a few IgG should be produced even after the first immunization.
2. Could authors show the IGN-gamma production in the control group of mice that have been vaccinated with PBS+Alum?
3. Have you evaluated to see if there are spike -specific memory b cells in the spleen by flow cytometry?
Author Response
- Can authors explain why there is no IgG detected after the first immunization? I agree that the booster is necessary but at least a few IgG should be produced even after the first immunization.
Author’s response: No significant antibody titer after first immunization was observed, which could be due to low amount of antigen or adjuvant used. Further studies are needed to determine the minimal antigen dose for maximal immune response after immunization.
- Could authors show the IFN-gamma production in the control group of mice that have been vaccinated with PBS+Alum?
Author’s response: Table S2. The result of IFN-γ ELISpot assay was provided in the Supplementary Table S2
|
Group |
Vaccination |
Sample No. |
IFN-γ ELISpot (SFU/106Splenocyte) |
|||||
|
Negative |
Positive |
SARS-CoV-2-Pool#3 |
SARS-CoV-2-Pool#4 |
SARS-CoV-2-Pool#5 |
Total |
|||
|
1 |
PBS + Alum |
1 |
0 |
2496 |
0 |
0 |
0 |
0 |
|
2 |
0 |
2446 |
0 |
0 |
0 |
0 |
||
|
3 |
0 |
578 |
0 |
0 |
0 |
0 |
||
|
4 |
0 |
2726 |
0 |
0 |
0 |
0 |
||
|
5 |
0 |
2616 |
0 |
0 |
0 |
0 |
||
|
Mean ± SD |
0 ± 0 |
|||||||
|
2 |
10 µg SARS-CoV-2 S1-Fc + Alum |
1 |
0 |
2244 |
0 |
0 |
172 |
172 |
|
2 |
0 |
1952 |
22 |
0 |
0 |
22 |
||
|
3 |
0 |
0 |
0 |
0 |
2 |
2 |
||
|
4 |
0 |
2432 |
90 |
32 |
112 |
234 |
||
|
5 |
0 |
2268 |
40 |
138 |
54 |
232 |
||
|
Mean ± SD |
132.4 ± 112.9 |
|||||||
- Have you evaluated to see if there are spike -specific memory b cells in the spleen by flow cytometry?
Author’s response: We didn’t perform the flow cytometry analysis for assessing the B cell subsets in the present study.
Reviewer 3 Report
In the article “Plant-produced S1 Subunit Protein of SARS-CoV-2 Elicits Immunogenic Responses in Mice” the authors made a COVID-19 subunit vaccine by agroinfiltration, and they tested it in a mouse model. The results are promising, but the overall quality of the article doesn’t warrant it to be published in a journal with an impact factor of ~5.
The writing is severely lacking, an English language professional should rewrite the article. The choice of words and the sentence structure is wrong in multiple occasions. Therefore, the message can be misinterpreted.
The statistics section is missing from the Methods.
The sample size is too low for this kind of experiment.
The authors should do the viral challenge experiments too and publish them together with these results if they want to publish in high IF journal.
Author Response
- In the article “Plant-produced S1 Subunit Protein of SARS-CoV-2 Elicits Immunogenic Responses in Mice” the authors made a COVID-19 subunit vaccine by agroinfiltration, and they tested it in a mouse model. The results are promising, but the overall quality of the article doesn’t warrant it to be published in a journal with an impact factor of ~5.
Author’s response: We sincerely appreciate all valuable comments and suggestions, which helped us to improve the quality of our manuscript. The paper has been carefully revised for better clarification.
- The writing is severely lacking, an English language professional should rewrite the article. The choice of words and the sentence structure is wrong in multiple occasions. Therefore, the message can be misinterpreted.
Author’s response: The manuscript was edited for correct English language, grammar, punctuation, and phrasing as per suggestion.
- The statistics section is missing from the Methods.
Author’s response: All statistical analyses were performed by using GraphPad Prism 9.0. Statistical section has been included in methods as per suggestion (Line numbers 199-203)
- The sample size is too low for this kind of experiment.
Author’s response: We agree with the reviewer. Our aim in this study is to assess the immunogenicity of plant-produced S1 based subunit vaccine against SARS-CoV-2. Further optimization experiments to evaluate the optimal antigen-adjuvant dose is required to induce the high antibody titer.
- The authors should do the viral challenge experiments too and publish them together with these results if they want to publish in high IF journal.
Author’s response: We agree with the reviewer. In this study, we assessed the immunogenicity of plant-produced S1 based subunit vaccine against SARS-CoV-2. Further, we planned to do challenge experiments in hACE 2 mice, safety and toxicity studies in animals in order to evaluate the potential of this candidate vaccine.
Round 2
Reviewer 3 Report
The article is improved compared to the previous version, the authors addressed most of the issues. I don't have further questions.